# Label-Free and Redox Markers-Based Electrochemical Aptasensors for Aflatoxin M1 Detection

**DOI:** 10.3390/s18124218

**Published:** 2018-12-01

**Authors:** Stefanos Karapetis, Dimitrios Nikolelis, Tibor Hianik

**Affiliations:** 1Department of Nuclear Physics and Biophysics, Comenius University, Mlynska dolina F1, 842 48 Bratislava, Slovakia; stevekara@chem.uoa.gr; 2Laboratory of Inorganic & Analytical Chemistry, School of Chemical Engineering, Department of Chemical Sciences, National Technical University of Athens, 9 Iroon Polytechniou St., 157 80 Athens, Greece; 3Laboratory of Environmental Chemistry, Department of Chemistry, University of Athens, Panepistimiopolis-Kouponia, 157 71 Athens, Greece; dnikolel@chem.uoa.gr

**Keywords:** biosensors, aflatoxin M1, aptasensors, electrochemical impedance spectroscopy, differential pulse voltammetry

## Abstract

We performed a comparative analysis of the sensitivity of aptamer-based biosensors for detection mycotoxin aflatoxin M_1_ (AFM1) depending on the method of immobilization of DNA aptamers and method of the detection. Label-free electrochemical impedance spectroscopy (EIS) and differential pulse voltammetry (DPV) for ferrocene labeled neutravidin layers were used for this purpose. Amino-modified DNA aptamers have been immobilized at the surface of polyamidoamine dendrimers (PAMAM) of fourth generation (G4) or biotin-modified aptamers were immobilized at the neutravidin layer chemisorbed at gold surface. In the first case the limit of detection (LOD) has been determined as 8.47 ng/L. In the second approach the LOD was similar 8.62 ng/L, which is below of allowable limits of AFM1 in milk and milk products. The aptasensors were validated in a spiked milk samples with good recovery better than 78%. Comparative analysis of the sensitivity of immuno- and aptasensors was also performed and showed comparable sensitivity.

## 1. Introduction

One of the main toxic factors in dairy products is the concentration of mycotoxins, which are secondary metabolites produced by organisms of the fungus kingdom [1]. They can cause diseases and even deaths either in humans or in animals [2]. The term “mycotoxin” is usually intended for the toxic chemical products of fungi. The fungi consume organic matter where humidity and temperature are adequate. If the conditions are appropriate, fungi proliferate into colonies and mycotoxin levels become high. The reason for the production of mycotoxins is not yet known [3].

The major groups of mycotoxins are the aflatoxin, ochratoxin, citrinin, ergot alkaloids, patulin and fusarium. In this work we focused on development of biosensor for detection aflatoxin M1 (AFM1), which is important pollutant of dairy products. Aflatoxin M1 is the natural metabolite of aflatoxin B1 (AFB1). Its presence in feed and the subsequent exposure of lactating animals lead to the contamination of milk by its hydroxylated metabolite, AFM1 [4,5]. Therefore, fresh milk must be regularly checked for the concentration level of AFM1. If the concentration of AFM1 is above 0.05 μg/kg (0.15 nM), or 0.5 μg/kg (1.5 nM) (according to EU or USA regulations, respectively), this milk cannot be used in the human food chain. This limit is even lower (0.025 μg/kg or 0.08 nM) for infant milk, follow-on milk, and dietary foods for infants [6].

The commercially used methods for AFM1 determination are mostly based on high-performance liquid chromatography (HPLC) [7], which advantageously substituted the thin layer chromatography technique (TLC) [8]. Enzyme-Linked Immunosorbent assay (ELISA) has become also very popular for mycotoxin analysis, leading to the development of many commercially available kits, which are essentially based on competitive assays [8]. Although the abovementioned techniques allowing detection of AFM1 with the required sensitivity and selectivity, they are rather expensive, time consuming and require qualified staff.

Biosensor technology can overcome the abovementioned difficulties. For instance, some biosensors in contrast with ELISA allow label-free detection of the analyte. Most of the biosensors for aflatoxin detection reported so far use one of two types of receptors as recognition elements—antibodies and DNA aptamers. Immunosensors are based on the immobilization of specific monoclonal antibodies on various surfaces. DNA (or RNA) aptamers represent a new alternative to expensive and less stable antibodies. Aptamers are single stranded DNA or RNA that in solution folds into a 3D structure with a binding site specific to the analyte. Aptamers are produced in vitro by the Systematic Evolution of Ligands with Exponential enrichment (SELEX) method using affine chromatography separation of randomly synthesized sequences [9,10]. The specificity of aptamers to the analyte is comparable with those of antibodies, but aptamers are more stable, easier to immobilize on surfaces and easier to modify chemically.

The main problems of immunosensors are the elevated cost of antibodies and in the ethical issues, as immunoglobulins are produced through animal immunization. Among potential recognition element alternatives, aptamers have been proposed as very promising tools in biosensing applications owing to their many advantages such as cost effectiveness, flexibility, ease of modification, high stability, and compatibility with large-scale production [11]. 

The first electrochemical immunosensor for AFM1 detection was reported by Micheli et al. [12]. In this work the design of the sensor was adopted from ELISA, but AFM1 specific antibodies were immobilized directly on screen-printed electrodes. The secondary antibody conjugated with horse radish peroxidase (HRP) was used for chronoamperometric detection of AFM1 with a detection limit of 25 ppt (25 ng/L) and a working range between 30 and 160 ppt. This range was comparable with those of ELISA, but the detection time was shorter. Vig et al. described an impedimetric immunosensor based on gold-labeled antibodies. This sensor detected AFM1 in a concentration range between 15 and 1000 ng/L (45 pM–3 nM) [13]. A competitive immunoassay based on antibodies immobilized on a gold microelectrode array and horseradish peroxidase (HRP)-AFM1 conjugate has been reported by Parker et al. [14]. The assay allows detection of AFM1 with a LOD of 8 ng/L in a dynamic range of 10 to 100 ng/L.

More recently, an indirect competitive immunoassay has been elaborated with a screen-printed electrode array adapted with a standard 96-well microplate for the analysis of frozen and lyophilized milk [15]. Paniel et al. detected AFM1 with a sensor that was based on a competitive immunoassay using HRP-AFM1 conjugate as a tag [16]. The samples containing AFM1 were incubated with a fixed amount of antibody and HRP-AFM1 tracer until the system reached equilibrium. The sensor allowed detection of AFM1 with LOD 10 ng/L. Bacher et al. reported a label-free impedimetric immunosensor based on a silver wire electrode that allows the detection of AFM1 within the range 6.25–100 ng/L (18.75 pM–0.3 nM), with a LOD of 1 ng/L (3 pM) [17].

In the last years an immunochromatographic strip with monoclonal antibodies was described for sensitive AFM1 detection in pasteurized and powdered milk [18]. Later, an electrochemical immunosensor with capture antibodies immobilized on the Au screen-printed electrode and signaling antibodies conjugated with alkaline phosphatase was tested in spiked milk samples with a LOD of 37 ng/L [19]. Chalyan et al. developed a sensing device that consisted of four silicon oxynitride (SiON) micro-ring resonators that detected AFM1 with off-chip silicon photodetectors onto which either DNA aptamers or antigen binding fragments (Fab’) were immobilized [20]. The most reproducible signal was obtained for sensors based on a Fab’-functionalized surface with a LOD of 5 nM (1641 ng/L). Immunosensors for AFM1 detection have been recently reviewed [21,22,23,24].

The first electrochemical aptasensor for AFM1 detection was reported by Nguyen et al. [25]. The sensor design was based on an interdigitated electrode array (IDA) fabricated on a silicon substrate via a lithography technique. Silicon wafers were covered with a SiO_2_ layer. Chromium (Cr) and platinum (Pt) layers were sputtered on the top of the wafer. After electropolymerization with Fe_3_O_4_/PANi, the aptamers were immobilized on this layer and detection of AFM1 was performed by differential pulse voltammetry (DPV). The sensor allowed detection of AFM1 with LOD 1.98 ng/L in the range of 6–60 ng/L. AFM1 has been also detected by square wave voltammetry (SWV) using an aptamer-based biosensor designed with a streptavidin layer coated by biotinylated aptamers with a dynamic range of 1–10^5^ ppt [26]. Later an impedimetric aptasensor designed with a 4-carboxy- phenyldiazonium salt surface on screen-printed electrodes (SPCE) has been proposed by Istamboulie et al. [27]. The working range of this sensor was 2–150 ng/L with a LOD of 1.15 ng/L.

The latest works described an AFM1 sensitive aptasensor based on a glassy carbon electrode (GCE) covered with polymeric Neutral Red dye (NR) obtained by electropolymerization in the presence of a polycarboxylated pillar[5]arene derivative [28]. In this case the detection of AFM1 was performed using electrochemical impedance spectroscopy (EIS) in a dynamic range of 15–120 ng/L (LOD 0.5 ng/L). An optical label-free aptasensor using total internal reflection ellipsometry (TIRE) combined with a localized surface plasmon resonance (LSRP) phenomenon in nanostructured gold films immobilized with specific aptamers was reported [29]. The working range was 10 ng/L–100 μg/L and the LOD 10 ng/L. Jalalian et al. also reported a novel electrochemical aptasensor based on a hairpin-shaped aptamer, gold nanoparticles (Au-NPs) and a complementary strand of the aptamer (CS). The DPV allowed detecting AFM1 in the range of 2–600 ng/L with a LOD of 0.9 ng/L [30]. Recent achievements in electrochemical DNA sensors and available techniques have been reviewed by Fojta et al. [31].

Among alternative methods of detection, it is worth mentioning a biosensor based on a metal- supported bilayer lipid membrane (s-BLM) [32]. It has been shown that AFM1 affected hybridization of DNA at the surface of s-BLM, which was recorded as changes of the displacement current. The LOD obtained by this approach was 0.5 nM of AFM1 in a working range of 1.9–20.9 nM.

In aptasensors the sensitivity and dynamic range depend on the method of aptamer immobilization as well as on the signal detection method. However, a comparative analysis of the effects of the aptamer immobilization and the method of detection has not been reported yet. In this work we present electrochemical sensors based on two methods of aptamer immobilization on a surface composed either of dendrimers or neutravidin. Impedance spectroscopy (EIS) and differential pulse voltammetry (DPV) have been used for the detection of AFM1 with aptamers.

## 2. Materials and Methods

### 2.1. Reagents and Aptamers

Aflatoxin M1 (AFM1, m.w. 328.27 Da) from *Aspergillus flavus*, cystamine dihydrochloride 96% (Cys), sodium borohydride NaBH_4_, glutaraldehyde solution in H_2_O 25% (GA), poly(amido-amine) dendrimers generation 4.0, 10 wt.% in methanol (PAMAM G4), 1-ethyl-3-(3-dimethylaminopropyl) carbodiimide (EDC), N-hydroxysuccinimide (NHS), potassium hexacyanoferrate(II) trihydrate K_4_Fe(CN)_6_·3H_2_O, potassium ferricyanide(III) K_3_Fe(CN)_6_, ferrocene carboxylic acid, Fc-COOH, tablets for preparation of phosphate buffer saline (PBS) as well as other standard chemicals were of p.a. grade and purchased from Sigma-Aldrich, (Schnelldorf, Germany). Neutravidin (NA) was supplied by Biotech (Bratislava, Slovakia). Sulfuric acid 96% H_2_SO_4_, hydrogen peroxide 30% H_2_O_2_, and ethanol, p.a. grade were from Slavus (Bratislava, Slovakia).

We used following buffers: PBS (137 mM NaCl, 27 mM KCl, 10 mM Na_2_HPO_4_, 1.8 mM KH_2_PO_4_, pH = 7.4) and HEPES 10 mM pH 7.4.

Following aptamers were used in experiments: 21-mer DNA aptamer of following sequence: 5′-ACT GCT AGA GAT TTT CCA CAT-3′ (APT1) described by Nguyen et al. [25] and modified at the 5′ end by an amino group or by biotin at the 5′ end, but containing a dT_15_ spacer: 5′-TTT TTT TTT TTT TTT ACT GCT AGA GAT TTT CCA CAT-3′ (APT2) and 50-mer DNA aptamer modified by biotin at the 5′ end: 5′-GTT GGG CAC GTG TTG TCT CTC TGT GTC TCG TGC CCT TCG CTA GGC CCA CA-3′ (APT3) [20,33] that is characterized by high specificity to AFM1 (dissociation constant K_d_ = 10 nM). Aptamers were purchased from Eurogentec (Seraing, Belgium). For preparation of stock solution of aptamers, we used TE buffer (10 mM Tris-HCl, 1 mM EDTA, pH = 8) based on DNase- free water (Sigma-Aldrich Schnelldorf, Germany). The working solutions of aptamers were prepared by dilution of the abovementioned stock solution in a corresponding working buffer.

### 2.2. Preparation of Aptasensors

The aptasensors were prepared using gold electrodes of 2 mm diameter (CH Instruments, Austin, TX, USA). The electrodes were carefully cleaned according to the procedure described earlier [34,35]. The gold surface was polished using an electrode-polishing kit consisting of 1.0 and 0.3 μm alumina powder (CH Instruments) and then sonicated for 15 min in ethanol followed by 20 successive cycles of electrochemical cleaning in 1 M H_2_SO_4_ in the potential range from +0.2 to +1.5 V vs. Ag/AgCl reference electrode with a scan rate of 100 mV/s. After that the electrodes were properly rinsed again in deionized water, ethanol and dried under nitrogen and the immobilization started immediately.

#### 2.2.1. Aptasensors Based on Immobilization Aptamers at Dendrimer Modified Surface

The preparation of an aptasensor on a dendrimer surface was adopted from our previous work [35]. Briefly, after cleaning the gold surface, the electrode was immersed in 0.1 M aqueous solution of cystamine for 2 h. This resulted in formation of a self-assembled cystamine layer due to chemisorption. After careful cleaning of the surface in deionised water it was immersed in 5% glutaraldehyde (GA) in water for 1 h, rinsed with PBS and incubated with 70 μM PAMAM G4 dendrimers in PBS buffer for 5 h. The unreacted groups of GA were blocked by immersion of the electrode in 5 mM NaBH_4_ during 30 min. The electrode has been then rinsed in PBS and again immersed into 5% GA solution for 1 h. This surface was ready for immobilization of amino-modified APT1. For this purpose, the electrode was immersed in 1 μM APT1 dissolved in PBS for 16 h and then again in 5 mM NaBH_4_ for 30 min. After rinsing the surface by PBS the aptasensor was used for AFM1 detection.

The sensing surface scheme is presented in Figure 1A. The AFM1 detection was performed using electrochemical impedance spectroscopy in the presence of 5 mM redox couple [Fe(CN)_6_]^3−/4−^. It is based on determination of the charge transfer resistance R_ct_ that changes in the presence of an analyte. This is known approach that was used in previous aptasensor research (see [36] and Section 2.3 for details).

#### 2.2.2. Aptasensors Based on Biotinylated Aptamers Immobilized on Neutravidin Layers

These aptasensors were prepared using DNA aptamers (APT2 or APT3) modified at the 5′ end with biotin and immobilized on a neutravidin layer chemisorbed at the surface of gold electrodes. This is based on the strong affinity of biotin for neutravidin. In order to detect AFM1 electrochemically the neutravidin layer was modified by ferrocenecarboxylic acid (Fc-COOH), that displays well-resolved redox signals. The scheme of the sensing layer is presented in Figure 1B. The aptasensor has been prepared using the following steps. First a clean gold electrode was immersed in neutravidin dissolved in deionised water at a concentration of 125 μg/mL for 15 min. This is the common way to prepare a stable neutravidin layer on a gold surface [37]. After rinsing in deionised water the electrode with the neutravidin layer was immersed in a solution containing activated Fc-COOH for 2 h. The activation has been performed according to the procedure described in [38]. Briefly, 20 nM Fc-COOH has been added into the mixture of 1 mM EDC and 5 mM NHS for 15 min. Activated Fc-COOH was prepared freshly before its immobilization at neutravidin layer. Addition of activated Fc-COOH onto the neutravidin layer resulted in its strong binding to NA molecules. After rinsing the ferrocene-neutravidin layer with deionized water and PBS the electrode was immersed in a 1 μM solution of biotinylated aptamer for 30 min. After rinsing with PBS, the sensor was ready for AFM1 detection. The detection was based on measurement of the ferrocene redox current. It can be expected that in the presence of AFM1 the environment close to the sensing surface changes which may affect the redox current. This approach has been used in this work for the first time, although similar principle was applied in our recent work [28]. In this case, however instead of Fc we used redox properties of electropolymerized layers formed by Neutral Red. In all cases the detection of AFM1 was accomplished by immersion of the sensor into the AFM1 solution in a concentration range of 15–120 ng/L. The sensor was incubated with corresponding concentration of AFM1 during 1 h at ambient temperature (approx. 22 °C).

### 2.3. Electrochemical Measurements

#### 2.3.1. Instrumentation

The electrical parameters of aptasensors were determined by an AUTOLAB PGSTAT12 potentiostat-galvanostat equipped with a FRA impedance module (Metrohm Autolab b.v., Herisau, Switzerland) or a CHI 440 potentiostat (CH Instruments) in a 8 mL Teflon cell using three electrode configuration consisting in working gold electrode (diameter 2 mm), Ag/AgCl reference electrode and Pt wire as auxiliary electrode. All electrodes were from CH Instruments. The sensor response following addition of AFM1 was measured by electrochemical impedance spectroscopy (EIS) and differential pulse voltammetry (DPV) with the following parameters. EIS: frequency range from 0.1 Hz to 100 kHz by applying 5 mV voltage amplitude and DC potential 0.22 V. EIS experiments were performed in PBS containing 5 mM (1:1) [Fe(CN)_6_]^3−/4−^. Data from Nyquist plots were fitted according to Nova software version 1.7 (Metrohm Autolab, b.v). Randle’s equivalent circuit (see inset in Figure 2) has been used in the analysis of the charge transfer resistance, R_ct_. In the case of DPV detection, the following parameters were used: potential range from 0 to +0.5 V with a resting time of 2 s.

#### 2.3.2. Validation of the Aptasensors in Milk Samples

The milk samples were prepared according to the procedure described in our recent paper [28]. A sample of UHT cow milk, 3.5% fat, obtained from local supermarket was first spiked with a certain amount of AFM1 in a concentration range of 15–120 ng/L and then thermostated at 40 °C. After that, the sample was diluted with methanol in the 3:1 *v*/*v* ratio and centrifuged at 5000 rpm for 5 min. The supernatant was diluted to 1:10 *v*/*v* ratio with the PBS. The aptasensor was incubated in the milk sample for 60 min and then its signal was recorded. The sensor recovery was calculated according to: [(ΔR_ct_/R_cto_)_milk_/(ΔR_ct_/R_cto_)_PBS_] × 100%, where (ΔR_ct_/R_cto_)_PBS_ are changes of charge transfer resistance at certain concentration of AFM1 in a buffer and (ΔR_ct_/R_cto_)_milk_ those in a milk, respectively.

## 3. Results and Discussion

### 3.1. Determination of AFM1 by Aptasensors Depending on Aptamer Immobilization and Detection

In this part we compare the response of aptasensors depending on the method of immobilization and detection. As a first we used aptamers APT1 modified at the 5′ end by an amino group. The aptamers were immobilized at the surface of PAMAM dendrimers (see Figure 1A). Because AFM1 is not electroactive, as a convenient tool for detection of AFM1 we selected electrochemical impedance spectroscopy (EIS) at the presence of the [Fe(CN)_6_]^3−/4−^ redox couple. At a formal potential of approx. 0.22 V vs. Ag/AgCl reference electrode (determined by CV) the charge transfer between the redox couple and the electrode surface is maximal. The changes at the sensing surface such as binding of AFM1 can affect this charge transfer. Therefore, detecting the changes in the charge transfer resistance, R_ct_, allows analysis of the interaction of AFM1 with the sensing surface. In addition, EIS allows analysis of all steps of the sensing surface preparation. This is demonstrated on Figure 2 where the corresponding Nyquist plot is presented. It can be seen that the plot consists of semicircles and linear parts which depend on the diffusion of the redox couple to the sensing surface. The Nyquist plot can be characterized by its Randles equivalent circuit (inset in Figure 2). The diameter of the semicircles is proportional to the R_ct_ values. The straight line corresponding to cystamine layer chemisorbed at the gold surface is due to the high conductivity of this structure, which is caused by a more rapid diffusion of redox markers. Also the PAMAM adlayer revealed these properties partially due to its positive charge that make the diffusion of the redox probe close to the electrode surface easier. Immobilization of aptamers resulted in an increase of semicircle diameter. This is due to the fact that DNA aptamers are negatively charged. As a result, the redox couple is repulsed from the electrode surface which increases the R_ct_ values. This agrees well with our previous work [39].

Figure 3A shows a Nyquist plot following stepwise incubation of the sensing surface with an increased concentration of AFM1. It can be seen that addition of AFM1 to the sensor surface resulted in an increase of the diameters of semicircles. This can be due to establishment of a barrier that partially blocks the diffusion of the redox couple from the solution to the electrode surface. Using the NOVA software (Metrohm Autolab b.v.) we fitted the Nyquist plot using the Randles equivalent circuit (lines in Figure 3A) and determined the charge transfer resistance with (R_ct_) and without (R_ct0_) AFM1. The plot of the relative changes of charge transfer resistance (ΔR_ct_/R_cto_) vs. AFM1 concentration is presented in Figure 3B.

It can be seen that with increasing concentration of AFM1 the relative changes of the resistance increase. The plot of ΔR_ct_/R_cto_ vs. AFM1 concentration is close to linear up to 60 ng/L of AFM1, then saturation started. Using the signal to noise ratio rule (S/N = 3) we determined LOD as 8.47 ng/L (see also Table 1). This is below the allowable limit established by EU legislation, so the sensor can be used in practical applications. The validation of this sensor in a spiked milk samples is presented below in the Section 3.3.

The aptasensor presented above and the EIS method of analysis can be considered as label-free, because does not require any labeling of aptamers or the sensing surface. This is advantageous in respect of practical applications for which the low cost is among the priorities considering the rather large number of milk samples that should be analyzed.

In order to compare the sensitivity of AFM1 detection using label-free and label-based detection, we applied the following approach: we prepared an aptasensor using DNA aptamers (APT2 or APT3) modified at the 5′ end by biotin. The aptamers were immobilized on the surface of gold electrodes with a chemisorbed neutravidin layer. Prior to aptamer immobilization the neutravidin molecules were modified by Fc-COOH according to the procedure described above in Section 2.2.2. This modification allows detection of the changes at the sensing surface by monitoring the redox properties of ferrocene (Fc). This is possible by using a DPV method. We can expect that binding of AFM1 will cause conformational changes of the aptamer and as a result the redox current can also be affected. This principle has been used in our earlier work, but Fc has been immobilized at the surface of dendrimers attached to the multiwalled carbon nanotubes [40]. For this purpose, a special Fc-linker has been synthesized. This approach gives us also possibility to compare the sensitivity of the sensors prepared from two different aptamers. The DPV for the sensor based on APT2 is presented in Figure 4A and those for APT3 in Figure 4B. It can be seen that the amplitude of the current decreases in both cases with increasing AFM1 concentration. Considering the possible instability of Fc during long term experiments and during CV cycling we also performed 30 CV scans for the aptasensor composed of APT2 on a neutravidin layer modified by Fc both in PBS and HEPES. In both cases only negligible changes of CV have been observed (results are not shown). Thus, the decrease of the peak current cannot be caused by instability of Fc. In order to compare the sensitivity of the aptasensors based on APT2 and APT3, we constructed a plot of relative changes of the peak current ΔI/I_0_ vs. AFM1 concentration (Figure 5). Similarly, like above we determined the LOD for both sensors (8.52 and 8.64 ng/L for APT2 and APT3, respectively). Thus, the sensitivity of AFM1 detection by both aptamers does not differ significantly and is close to the values obtained based on the label-free aptasensor presented above using APT1 and the EIS method of detection. We should also mention that the sensitivity of Fc-based detection by APT3 is much higher in comparison with optical detection as reported in [20].

### 3.2. Comparison of the Sensitivity of Immuno- and Aptasensors in AFM1 Detection

As we mentioned above the aptasensors developed so far detect AFM1 with sufficient sensitivity comparable with that of immunosensors. However, aptasensors are more stable and cheaper in comparison with immunosensors. In addition, aptasensors are in principle reusable. Because the interaction of an analyte with the aptamer is based on electrostatic or Van der Walls interactions, the regeneration of the aptasensor is possible by immersion of the sensor in a solution of high ionic strength, for example 2 M NaCl, sodium dodecyl sulfate (SDS) or 0.2 M glycine-HCl solution. In particular, we proved this in our previous work on the detection of aflatoxin B1 using a dendrimer immobilization platform [35]. The sensor can be regenerated in 0.2 M glycine-HCl at least three times, without any significant lost of the sensitivity, which reduces the cost for its preparation. Similar aptasensor regeneration results were reported in [26]. In this case 10% SDS was used as regeneration agent. In contrast, immunosensors based on antibodies cannot be regenerated mostly due to the strong binding of the analyte to the antibody. Application of regeneration methods that are suitable for aptasensors resulted in irreversible changes in the antibodies that lost their binding affinity [41].

Table 1 compares the basic properties of immunosensors, DNA sensors and aptasensors for detection of AFM1 published so far. 

It can be seen that the sensitivity of aptasensors is comparable with that of immunosensors. The only exception is work by Chalyan et al. [20] in which an immunosensor based on an AFM1-selective Fab’ fragment revealed higher sensitivity in comparison with those based on aptamers analyzed by the micro-ring resonator method. However, the LOD of this immunosensor (1641 ng/L) is not sufficient for practical applications. Among so far published AFM1-sensitive aptasensors the highest sensitivity is revealed by those based on immobilization of aptamers on pillar[5]arene Neutral Red layers (LOD 0.5 ng/L) [28]. This sensor was label-free because does not require labeling of the aptamers. It has been successfully validated in various milk samples. A label-free aptasensor based on immobilization of an electroactive Fe_3_O_4_/PANi interface also revealed a rather good LOD of 1.98 ng/L. The only disadvantage of this sensor was its relatively low dynamic range (up to 60 ng/L), but the fabrication of this sensor as well as detection using DPV method was relatively easy. Unfortunately, the sensor has not been validated in real milk samples. A similar LOD (1 ng/L) has been reported in [26]. This electrochemical sensor was based on immobilization of biotinylated aptamers on streptavidin layers. For detection by square wave voltammetry (SWV), a 5 mM K_3_[Fe(CN)_6_] redox probe has been used. The sensor provided a surprisingly high dynamic range of 1–10^5^ ng/L. The advantage of this sensor was possibility of its regeneration in 10% SDS. However, validation in milk samples was missing. A rather simple aptasensor preparation method was presented in [27] using carbon screen-printed electrodes onto which amino-modified aptamers were immobilized by covalent binding. The sensor revealed good sensitivity (LOD 1.15 ng/L) and dynamic range. The sensor also revealed good stability and was validated in real milk samples with recoveries between 99–111%. The sensor can also be considered as label-free due to the application of the EIS method in the presence of 1 mM [Fe(CN)_6_]^3−/4−^ redox couple. Rather good sensitivity (LOD 0.9 ng/L) has been obtained in recent work by Jalalian et al. [30] using hairpin-shaped aptamers. The detection is based on conformational changes of the aptamers at the presence of AFM1 with subsequent hybridization of aptamers with added complementary DNA strands immobilized on gold nanoparticles. The only disadvantage of this assay is a more complicated scheme that requires both DNA-modified nanoparticles as well as methylene blue as a redox probe. This sensor has also been validated in real milk samples with recoveries of 91.3 to 96.5%. Finally, the optical sensor reported in [29] revealed also good LOD (10 ng/L). It was relatively easy to fabricate. The only disadvantage is that for detection of AFM1 more sophisticated methods of total internal reflection ellipsometry (TIRE) combined with localized surface plasmon resonance (LSPR) are required. The sensor, however, has not been validated in milk samples. The sensor presented in our work based on label-free EIS detection and those based on Fc-modified neutravidin layers are of comparable sensitivity and dynamic range, therefore they can be used in practical applications. We should especially point out the new method of AFM1 detection based on biotinylated aptamers immobilized at chemisorbed neutravidin layer modified by Fc. This aptasensor can be easily prepared and detection of AFM1 can be performed by DPV or CV methods that are not difficult to handle, so can be used even in remote milk laboratories. The preparation of this sensor is relatively rapid and requires approx. 3 h, which is much faster in comparison with those published so far. In addition, due to availability on the market of low cost potentiostats this detection is also cost effective.

### 3.3. Validation of the Biosensors in Spiked Milk Samples

Considering the advantages of label-free detection of AFM1 using aptasensors based on immobilization of aptamers on a dendrimer surface, we used these aptasensors for validation in a real milk sample. The samples were prepared according to the procedure described in Section 2.3.2. The experiments in a spiked milk samples were performed as follows: first the aptasensor was incubated in a milk without AFM1 during 1 h. After washing the surface in a buffer, it has been immersed in PBS containing 5 mM (1:1) [Fe(CN)_6_]^3−/4−^ redox probe and the R_ct_ value has been determined. The same procedure has then been performed but with milk samples spiked with certain concentrations of AFM1 in a range from 15 to 120 ng/L. Figure 6 compares the changes of charge transfer resistance vs. AFM1 concentration in buffer and in milk. It can be seen that there are only relatively small deviations between the values that evidence a good sensor recovery.

This is also demonstrated in Table 2 where the sensor recovery is calculated. The recovery varied between 78.04 to 106.25% which may be due to certain influence of milk proteins on the properties of the sensing surface, such as non-specific binding that can partially block the diffusion of the redox probe to the sensing surface.

## 4. Conclusions

Two aptasensors for the sensitive determination of aflatoxin M1 (AFM1) in aqueous solutions and in milk samples were developed and compared in respect of aptamer immobilization and method of detection. For the first time we report aptasensors for the detection of AFM1 based on aptamer immobilization on a dendrimer layers as well as on neutravidin layers modified by Fc-COOH. Comparison of label-free EIS based biosensor with those utilizing Fc-labelled neutravidin revealed a similar limit of detection (LOD), which is below the allowable contamination of the milk and milk products by AFM1. The aptasensor based on dendrimer layers has been validated in spiked milk samples and revealed recoveries between 78.04–106.25%. The possible deviation can be due to effect of milk proteins on the sensing layer properties. This will require further efforts to overcome this influence.

## Figures and Tables

**Figure 1 sensors-18-04218-f001:**
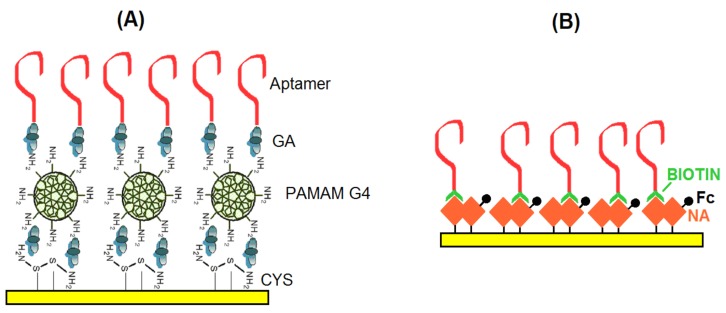
The schemes of aptasensors: (**A**) aptamers are immobilized at the surface of PAMAM G4 dendrimers. (**B**) biotinylated aptamers are immobilized at the neutravidin (NA) layer modified by ferrocene carboxylic acid (Fc). GA—glutaraldehyde, CYS—cystamine.

**Figure 2 sensors-18-04218-f002:**
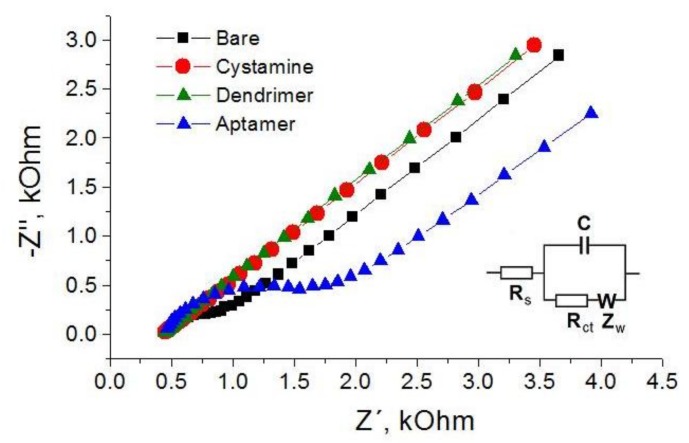
Nyquist plots corresponding to the main steps of aptasensor preparation from bare gold surface to cystamine layer, dendrimer adlayer and finally after immobilization of aptamers (see the legend). Inset represent Randles equivalent circuit that models electrical properties of the layers. R_s_ and R_ct_ are the electrolyte and charge transfer resistances, respectively. Z_w_ is the Warburg impedance resulting from the diffusion of the redox probe and C is the capacitance of the electrode surface/solution interface. Experiments were performed in the working phosphate buffer containing 5 mM (1:1) [Fe(CN)_6_]^3−/4−^ as a redox probe.

**Figure 3 sensors-18-04218-f003:**
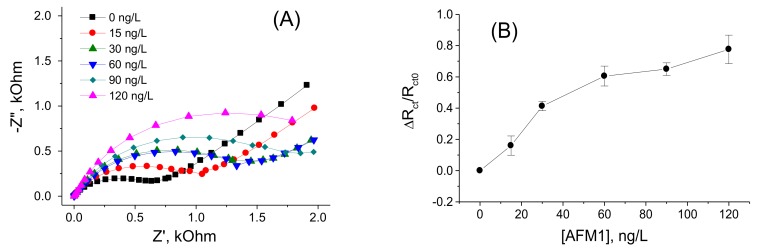
(**A**) Nyquist plots corresponding to the aptasensor without AFM1 and after incubation with various AFM1 concentrations (see the legend). (**B**) plot of the relative changes of R_ct_ values vs. concentration of AFM1 (ΔR_ct_/R_cto_ = (R_ct_ − R_cto_)/R_cto_, where R_cto_, R_ct_ are charge transfer resistances without and with certain concentration of AFM1, respectively). Results represent mean ± SD from 3 independent experiments. Experiments were performed in working phosphate buffer containing 5 mM (1:1) [Fe(CN)_6_]^3−/4−^ as a redox probe.

**Figure 4 sensors-18-04218-f004:**
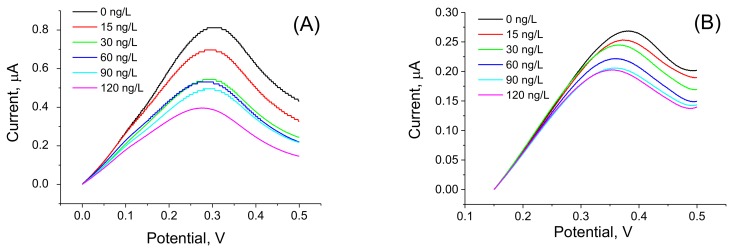
DPV of aptasensors based on biotinylated aptamers immobilized at Fc-modified neutravidin layer. (**A**) APT2 (in PBS), (**B**) APT3 (in HEPES).

**Figure 5 sensors-18-04218-f005:**
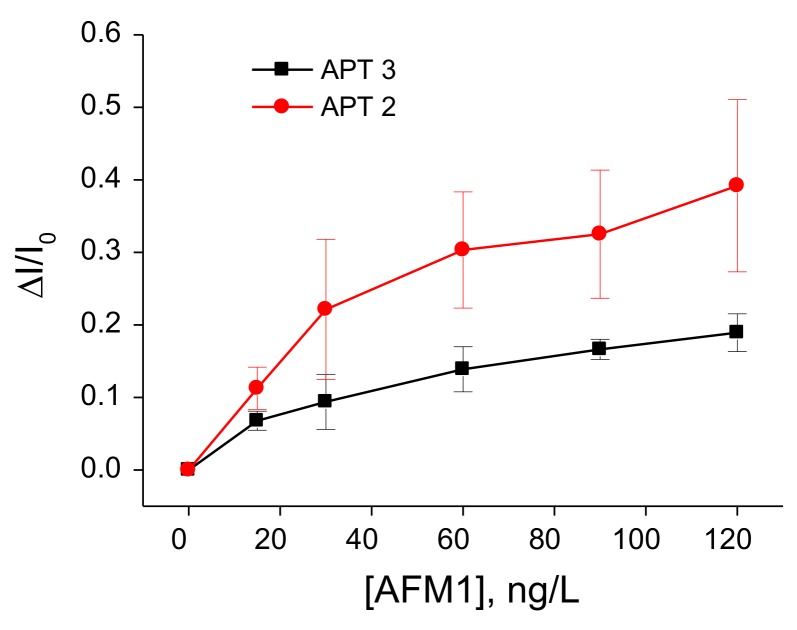
Plot of the relative changes of peak current ΔI/I_0_ vs. concentration of AFM1 constructed based on the DPV presented on Figure 4. (ΔI/I_0_ = (I − I_0_)/I_0_, where I is the peak current at certain concentration of AFM1 and I_0_ those without AFM1). Results represent mean ± SD obtained from 3 independent experiments in each series.

**Figure 6 sensors-18-04218-f006:**
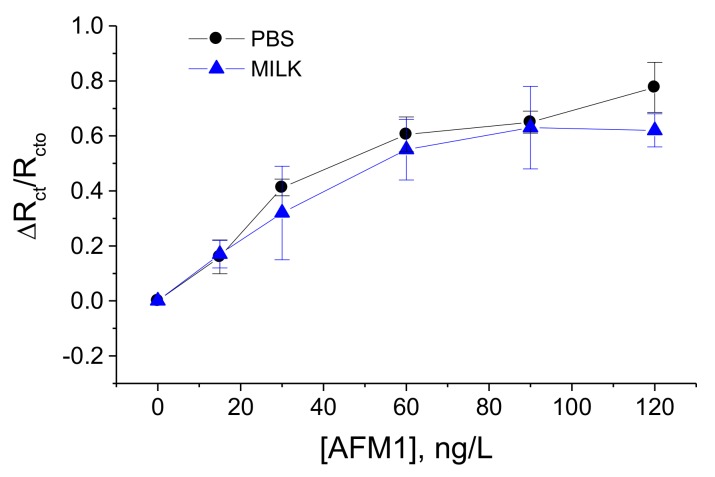
Plot of the relative changes of R_ct_ values vs. concentration of AFM1 (ΔR_ct_/R_cto_ = (R_ct_ − R_cto_)/R_cto_, where R_cto_, R_ct_ are charge transfer resistances without and with certain concentration of AFM1, respectively) in a PBS and in a spiked milk samples (see the legend). Results represent mean ± SD from 3 independent experiments in each series. Experiments were performed at presence of 5 mM (1:1) [Fe(CN)_6_]^3−/4−^ as a redox probe.

**Table 1 sensors-18-04218-t001:** Comparison of the LOD values of electrochemical aptasensors for AFM1 determination.

Sensor Preparation	Method of Detection	Dynamic Range, ng/L	LOD, ng/L	Reference
**Immunosensors**				
Immobilization of the specific antibodies on the screen-printed electrode	ELISA	30–160	25	[12]
An amperometric immunosensor based on the gold-labeled antibodies immobilized at screen-printed electrodes	ELISA	15–1000	15	[13]
A screen-printed electrode array adapted with a standard 96-well microplate	ELISA	5–250	1	[15]
A sensor based on a competitive immunoassay using horseradish peroxidase (HRP)	Chronoamperometry	10–500	10	[16]
A label-free impedimetric immunosensor based on silver wire electrode	ELISA	1–100	1	[17]
Immunochromatographic strip with immobilized AFB1–bovine serum albumin as the immobilized antigen and anti-AFM1 antibody labeled with gold nanoparticles as tracers	ELISA	-	200	[18]
An electrochemical immunosensor with capture antibodies immobilized on the gold screen-printed electrode. Competitive assay	DPV	-	37	[19]
Microelectrode array immunosensor with antibodies immobilized by cross-linking with 1,4-phenylene diisothiocyanate.	ELISA	1–100	8	[42]
Antigen-binding fragments (Fab’) immobilized on silicon oxynitride micro ring resonators	MRR	-	1641	[20]
**DNA Sensors**				
Metal-supported bilayer lipid membranes (s-BLMs)	Amperometry	0.5–6572	157	[31]
**Aptasensors**				
Aptasensor with electrochemical Fe_3_O_4_/PANi interface	DPV	6–60	1.98	[25]
Aptasensor based on biotin-modified aptamer at streptavidin layer on a screen-printed electrode	CV, SWV	1–10^5^	1	[26]
Hexaethyleneglycol-modified aptamers immobilized on a carbon screen-printed electrode	CV, EIS	2-150	1.15	[27]
Neutral Red electropolymerized film modified by pillar[5]arene	EIS	5–120	0.5	[28]
Optical label-free. Aptamers immobilized on nanostructured Au films	TIRE with LSRP	10–10^5^	10	[29]
Hairpin-shaped aptamer immobilized on gold nanoparticles. Methylene blue as a redox probe	DPV	2–600	0.9	[30]
Aptamers immobilized at PAMAM dendrimers	EIS	15–120	8.47	This work
Biotinylated aptamers immobilized at neutravidin layer modified by ferrocene	DPV	15–120	8.52	This work

CV—cyclic voltammetry; DPV—differential pulse voltammetry; EIS—electrochemical impedance spectroscopy; ELISA—Enzyme linked immunosorbent assay; LSRP—localized surface plasmon resonance; MRR—optical microring resonator; SWV—square wave voltammetry; TIRE—total internal reflection ellipsometry.

**Table 2 sensors-18-04218-t002:** Comparison of the EIS response of the aptasensor in a PBS and in spiked milk samples. Recovery was calculated as: [(ΔR_ct_/R_cto_)_milk_/(ΔR_ct_/R_cto_)_PBS_] × 100%.

Concentration of AFM1, ng/L	Buffer ΔR_ct_/R_cto_	Milk ΔR_ct_/R_cto_	Recovery, %
15	0.16	0.17	106.25%
30	0.41	0.32	78.04%
60	0.61	0.55	90.16%
90	0.65	0.63	97.00%
120	0.78	0.62	79.5%

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
