# Peer review of "Label-Free and Redox Markers-Based Electrochemical Aptasensors for Aflatoxin M1 Detection"

_sensors, 2018, doi:10.3390/s18124218_

Reviewer 1 Report

This very interesting research paper is devoted to the development of a new assay for sensitive determination of aflatoxin M1. The scientific quality of the manuscript and its importance for the field of bioanalytical chemistry and electrochemistry are undoubted. I really appreciate the amount of work performed during this research. There are only few points which could be improved during revision of the manuscript:

1.      In the Introduction, following review, tightly connected with the manuscript topic, could be also cited:

Fojta M., Daňhel A., Havran L., Vyskočil V.: Recent Progress in Electrochemical Sensors and Assays for DNA Damage and Repair. TrAC Trends in Analytical Chemistry 2016, 79, 160–167.

2.      Page 2, line 70: A dot is missing at the end of the sentence.

3.      Page 2, line 87: “et.al” should be “et al.”.

4.      Page 2, line 89: “–” should be “-”.

5.      At many places, “Fe (CN)” should be “Fe(CN)”.

6.      In caption to Figure 5: number of measurements (n), from which the error bars were constructed, should be mentioned.

7.      Sometimes, the terms “screen printed” and “screen-printed” are used and should be unified.

Author Response

We are grateful to this reviewer for most useful comments that allowed to improve manuscript. All changes made in the revised manuscript are highlighted by yellow.

Comment:

1. In the Introduction, following review, tightly connected with the manuscript topic, could be also cited:

Fojta M., Daňhel A., Havran L., Vyskočil V.: Recent Progress in Electrochemical Sensors and Assays for DNA Damage and Repair. TrAC Trends in Analytical Chemistry 2016, 79, 160–167.

Response:

We agree with reviewer and recommended publication has been included in the Introduction and in the list of references

Comments:

2.      Page 2, line 70: A dot is missing at the end of the sentence.

3.      Page 2, line 87: “et.al” should be “et al.”.

4.      Page 2, line 89: “–” should be “-”.

5.      At many places, “Fe (CN)” should be “Fe(CN)”.

Response:

The misprints were corrected as suggested

Comment:

6.      In caption to Figure 5: number of measurements (n), from which the error bars were constructed, should be mentioned.

Response:

The following addition was made in the caption to Figure 5: "Results represent mean±SD obtained from 3 independent experiments in ach series."

Comment:

7.      Sometimes, the terms “screen printed” and “screen-printed” are used and should be unified.

Response:

We agree with this comment and term "screen-printed" has been used uniformly in the text.

Reviewer 2 Report

These are the comments raised by the reviewer to improve the quality of current work.

1. It would be better to summarize the previous studies reporting the detection of AFM1 and address their limitations, rather than listing all the details such as detection range and LOD. 

2. Critical issue: Why did the authors use EIS for APT1 and DPV for APT2 and APT3? The detection techniques should be consistent to prove the superiority of the developed platforms over other platforms previously reported. Please explain.

3. Figure 4: It would be better to show wider potential range. 

4. The authors should specify which aptasensor was used for the real milk samples.

5. Is the aptasensor reusable? This is important for cost-effectiveness of the developed sensing platform. Please explain.

6. Following sentences are recommended to be corrected.

- “In the case of DVP following parameters were used: potential range from -0.2 to +0.7 V at quiet time of 2s” (Page 6, Line 232) should be corrected as “In the case of DPV detection, following parameters were used: potential range from -0.2 to +0.7 V at quiet time of 2s”.

Author Response

We are grateful to this reviewer for most useful comments that allowed to improve manuscript. All changes made in the revised manuscript are highlighted by yellow.

Comment:

1. It would be better to summarize the previous studies reporting the detection of AFM1 and address their limitations, rather than listing all the details such as detection range and LOD.

Response:

We agree with this comment and summarized previous studies of AFM1 in the section 3.2.

Comment:

2. Critical issue: Why did the authors use EIS for APT1 and DPV for APT2 and APT3? The detection techniques should be consistent to prove the superiority of the developed platforms over other platforms previously reported. Please explain.

Response:

The different methods of detection have been used due to different sensor design. In the case of APT1 the sensor surface was electrochemically not active. Therefore EIS at presence of redox couple was only possible method for detection of aptamer-AFM1 interactions.  APT2 and APT3 have been immobilized at the neutravidin layer modified by ferrocene (Fc). Thus, in this case the sensing surface was electrochemically active and it has been possible to use redox properties of Fc for detection the aptamer-AFM1 interactions. Interestingly, that in both cases the limit of detection was similar.

Response

3. Figure 4: It would be better to show wider potential range.

Comment:

The potential rage has been selected in order to detect in good resolution the peak current.

We did not extend the potential range to higher voltages in order to provide more soft conditions that will not affect the properties of sensing layer.

Comment:

4. The authors should specify which aptasensor was used for the real milk samples.

Response:

The aptasenor based dendrimer immobilisation platform with APT1 was applied for validation this sensor in the milk samples. This is mentioned in the text (section 3.3) as follows:

"Considering advantage of label-free detection of AFM1 using aptasensors based on immobilization aptamers at the dendrimer surface, we used these aptasensors for validation in a picked milk sample".

Comment:

5. Is the aptasensor reusable? This is important for cost-effectiveness of the developed sensing platform. Please explain.

Response:

Yes, the aptasensors are in principle reusable. Because the interaction of analyte with the aptamer is based on the electrostatic or Van der Walls interactions, the regeneration of the aptasensor is possible by immersion of the sensor into the solution of high ionic strength, for example 2 M NaCl, sodium dodecyl sulfate (SDS) or in 0.2 M glycine-HCl solution. In particularly, we approved this in our previous work for detection of aflatoxin B1 using dendrimer immobilization platform [35]. The sensor can be regenerated in 0.2 M glycine-HCl at least 3 times, without significant lost of the sensitivity, which reduce cost for its preparation. Similar results in aptasensor regeneration was reported in Ref. [26]. In this case 10 % SDS has been used as regeneration agent. In contrast, immunosensors based on antibodies can not be regenerated mostly due to strong binding of analyte to the antibody. Application of the method of regeneration that are suitable for aptasensor resulted in irreversible changes in antibodies that lost of the binding affinity [41].

We added this explanation in the revised manuscript (section 3.2).

Comment:

6. Following sentences are recommended to be corrected.

- “In the case of DVP following parameters were used: potential range from -0.2 to +0.7 V at quiet time of 2s” (Page 6, Line 232) should be corrected as “In the case of DPV detection, following parameters were used: potential range from -0.2 to +0.7 V at quiet time of 2s”.

Response:

We agree with this comment and the sentence has been corrected according to the reviewer's suggestion.